# A Distributed Architecture for Secure Delegated Quantum Computation

**DOI:** 10.3390/e24060794

**Published:** 2022-06-07

**Authors:** Shuquan Ma, Changhua Zhu, Dongxiao Quan, Min Nie

**Affiliations:** 1State Key Laboratory of Integrated Services Networks, Xidian University, Xi’an 710071, China; msqloveslife@outlook.com (S.M.); dxquan@xidian.edu.cn (D.Q.); 2Collaborative Innovation Center of Quantum Information of Shaanxi Province, Xidian University, Xi’an 710071, China; 3Shaanxi Key Laboratory of Information Communication Network and Security, Xi’an University of Posts & Telecommunications, Xi’an 710121, China; niemin@xupt.edu.cn; 4School of Communications and Information Engineering, Xi’an University of Posts & Telecommunications, Xi’an 710121, China

**Keywords:** quantum computation, secure delegated computation, distributed architecture

## Abstract

In this paper, we propose a distributed secure delegated quantum computation protocol, by which an almost classical client can delegate a (dk)-qubit quantum circuit to *d* quantum servers, where each server is equipped with a 2k-qubit register that is used to process only *k* qubits of the delegated quantum circuit. None of servers can learn any information about the input and output of the computation. The only requirement for the client is that he or she has ability to prepare four possible qubits in the state of $\left(|0\rangle +e^{i\theta }|1\rangle \right)/\sqrt{2}$, where θ∈{0,π/2,π,3π/2}. The only requirement for servers is that each pair of them share some entangled states $\left(|0\rangle |+\rangle +|1\rangle |-\rangle \right)/\sqrt{2}$ as ancillary qubits. Instead of assuming that all servers are interconnected directly by quantum channels, we introduce a third party in our protocol that is designed to distribute the entangled states between those servers. This would simplify the quantum network because the servers do not need to share a quantum channel. In the end, we show that our protocol can guarantee unconditional security of the computation under the situation where all servers, including the third party, are honest-but-curious and allowed to cooperate with each other.

## 1. Introduction

Quantum computing has been extensively studied from theory to practice [1,2]. It is widely accepted that noisy intermediate-scale quantum (NISQ) computers may be available in the coming decades [3]. However, the limited quantum memory of NISQ devices means that they may not have the capability to deal with large-scale quantum information processing. This is obviously a severe constraint, as many practical problems, e.g., *machine learning,* usually require immense memory overhead. A feasible way to overcome this obstacle is to utilize *distributed architecture* for quantum computations [4]. That is, using a group of small-scale quantum computers interconnected by classical and quantum networks to implement large-scale quantum computation tasks. However, considering the tremendous cost of building a quantum computer, it is not likely that ordinary consumers will be able to afford an NISQ computer in the foreseeable future. In fact, it is widely believed that the role of quantum computers is similar to today’s classical supercomputers, which means only a few organizations or enterprises can have quantum computers at their disposal. Thus, for ordinary customers, a better way to access quantum computers is to delegate their computations to the companies that offer quantum computing as cloud services. Indeed, this computation pattern has been applied in today’s Internet, e.g., IBM Quantum platform [5].

Delegated quantum computation is actually closely related to distributed quantum computation [4]. The client-to-server pattern in delegated computation naturally belongs to the category of distributed quantum computation. A class of delegated quantum computation protocols are constructed under the framework of measurement-based quantum computation (MBQC) [6,7,8], which is driven by a sequence of single-qubit measurements on some specific entangled state, where the entangled resource is also a basic module in the distributed quantum computation. Another class of delegated quantum computation protocols are obtained using the technique *quantum computing on encrypted data* (QCED) [9] or *quantum homomorphic encryption* (QHE) [10]. Although QCED and QHE are distinct concepts, the basic idea behind them is identical. Both of them use the *quantum one-time pad* to encrypt the input and output states but use different the methods to achieve the non-Clifford gates. Nevertheless, most schemes use the entangled states as the ancillary resources, for example [10,11,12].

Both distributed quantum computation and delegated quantum computation have been investigated broadly; see references [13,14,15,16,17,18,19,20,21] and [6,11,22,23,24,25,26,27,28], respectively. Typically, the distributed architecture for quantum computation makes use of photons as *flying qubits* between computational nodes, where each node is equipped with a quantum computer. The flying qubits are usually used to generate entangle states between distinct servers (i.e., nodes). By means of quantum entanglement, the non-local operations, such as controlled-NOT gate, can be done between two distant servers. Note that the quantum computer in each server is not necessarily an optical quantum computer; it can be made up of some other quantum system [29], such as ion traps or cloud atoms. Related experiments have been successfully demonstrated (see references [30,31]). Recently, researchers also investigated the possibility of simulating large-scale quantum systems in a hybrid quantum-classical manner [32]. That is, using a classical computer combined with a small quantum computer to simulate a large quantum computer [33]. However, the computational model considered in [32,33] is slightly different from the traditional model of circuit-based quantum computation. In this paper, we will not consider the method in [32], but rather the quantum entanglement to implement the non-local operation. In general, delegated quantum computation refers specifically to the *secure delegated quantum computation* (SDQC), which requires that no one except the client can obtain the right input and output of the computation. Typically, the client is required to have some basic quantum capacities, for example, preparing some single qubits or performing single-qubit measurements. In [34], the authors proposed a more rigorous SDQC protocol, which they called *universal blind quantum computation* (UBQC). The new protocol can guarantee that not only the input and output but also the computation itself, i.e., the algorithm, are unknown to the server. Although it seems that UBQC is more secure than SDQC, they are equivalent. That is, SDQC can be converted into UBQC [35]. As delegated quantum computation protocols effectively release the quantum resources in the client side, related experimental demonstrations have rapidly been implemented using the linear optics components (see References [9,25,36,37]).

Based on the above observations, in this paper we formally propose a distributed secure delegated quantum computation protocol that allows a half-classical client who can only prepare special single qubits to implement a large-scale quantum circuit on several quantum servers interconnected by entangled channels. Each server only has a limited quantum memory so that it can only compute a fraction of the delegated circuit. Moreover, during the computation, servers get nothing about the input and output of the computation. We also give a detailed security proof for our protocol. The rest of this paper is organized as follows. Section 2 introduces some basic preliminaries and notation. Section 3 presents the basic modules for delegated quantum computation. Section 4 gives the complete distributed delegated quantum computation protocol. Section 5 analyzes the security of our protocol. The last section discusses some remaining problems in our work.

## 2. Preliminaries and Notation

We assume that readers are familiar with the basics of quantum computation. In this work, we will use the following basic quantum gates: (1)$$\begin{array}{c} Z|s\rangle =e^{is\pi }|s\rangle , \end{array}$$(2)$$\begin{array}{c} X|s\rangle =|s⊕1\rangle , \end{array}$$(3)$$\begin{array}{c} H|s\rangle =\frac{1}{\sqrt{2}}\left(|0\rangle +e^{is\pi }|1\rangle \right), \end{array}$$(4)$$\begin{array}{c} P|s\rangle =e^{i\frac{s}{2}\pi }|s\rangle , \end{array}$$(5)$$\begin{array}{c} T|s\rangle =e^{i\frac{s}{4}\pi }|s\rangle , \end{array}$$(6)$$\begin{array}{c} CZ|s,t\rangle =e^{ist\pi }|s,t\rangle , \end{array}$$
where s,t∈{0,1} and $i=\sqrt{-1}$; *P* and *T* refer to the phase gate and the π/8 gate, respectively; and CZ denotes the controlled-*Z* gate. In order to analyze conveniently, we also introduce the *Z*-rotation operator defined as follows: (7)$$R_{z}\left(\alpha \right)=\left(\begin{array}{cc} e^{-i\frac{\alpha }{2}} & 0\\ 0 & e^{i\frac{\alpha }{2}} \end{array}\right),$$
where α∈[0,2π) is referred as the *rotation angle*. Regardless of the global phases, we can see that $Z\equiv R_{z}\left(\pi \right)$, $P\equiv R_{z}\left(\frac{\pi }{2}\right)$, and $T\equiv R_{z}\left(\frac{\pi }{4}\right)$. We use $|{+}_{\varphi }\rangle$ to denote the following single qubit: (8)$$|{+}_{\varphi }\rangle =\frac{|0\rangle +e^{i\varphi \pi }|1\rangle }{\sqrt{2}},$$
where we consider φ∈[0,2π). It is clear that, up to an unimportant global phase, $R_{z}\left(\alpha \right)|{+}_{\varphi }\rangle \equiv |{+}_{\left(\varphi +\alpha \right)}\rangle$. Thus, φ is also called as the *rotation angle*. By this definition, we can see that $|+\rangle =|{+}_{0}\rangle$ and $|-\rangle =|{+}_{\pi }\rangle$. Note that for any θ∈[0,2π) the states $|{+}_{\theta }\rangle$ and $|{+}_{\left(\theta +\pi \right)}\rangle$ comprise a basis, thus we can define a single-qubit measurement operator as follows: (9)$$M\left(\theta \right)=\sum_{s\in \{0,1\}}{\left(-1\right)}^{s}|{+}_{\left(\theta +s\pi \right)}\rangle \langle {+}_{\left(\theta +s\pi \right)}|,$$
where θ is referred as the *measurement angle* in this case, and s∈{0,1} denotes the classical measurement outcome. Specifically, s=0 if the post-measurement state is $|{+}_{\theta }\rangle$, otherwise s=1. Finally, in this work we will also use a special two-qubit entangled state defined as follows:(10)$$|H\rangle =\frac{|0\rangle |+\rangle +|1\rangle |-\rangle }{\sqrt{2}},$$
which can be prepared by applying a CZ gate on two qubits |+〉|+〉.

## 3. Secure Delegated Quantum Computation

In this work, the delegated quantum computation model we adopt is from [38], in which the authors improved the original QCED protocol [11] in two aspects. First, the quantum capacities of clients are further reduced. In theory, they only need to prepare the qubits $|{+}_{\varphi }\rangle$, where $\varphi \in \{0,\frac{\pi }{2},\pi ,\frac{3\pi }{2}\}$. Second, the security of the protocol can be still guaranteed even if some information is leaked to servers.

First of all, we specify that the client’s input is encoded in *X* basis. That is, encoding 0 and 1 as |+〉 and |−〉, respectively. Let $x=x_{1}x_{2}\cdots x_{n}\in {\left\{0,1\right\}}^{n}$ be the *n*-bit classical input string, then the corresponding encoded input state can be expressed as $|{+}_{x\pi }\rangle \equiv |{+}_{x_{1}\pi }\rangle |{+}_{x_{2}\pi }\rangle \cdots |{+}_{x_{n}\pi }\rangle$. For simplicity, we abbreviate $|{+}_{x\pi }\rangle$ as $|{+}_{x}\rangle$. The universal gate set we consider is U={X,Z,P,T,H,CZ}. Note that this gate set is not minimal because X,Z, and *P* can be obtained from {T,H}. Despite that, additional basic gates can effectively decrease the circuit complexity.

Now suppose the client’s input state is $|{+}_{x}\rangle$, where $x\in {\left\{0,1\right\}}^{n}$. In [38], the client uses the random operator $X_{i}^{a_{i}}Z_{i}^{b_{i}}P_{i}^{c_{i}}$ to encrypt each qubit $|{+}_{x_{i}}\rangle$, where $x_{i}\in \left\{0,1\right\}$, and $a_{i},b_{i},c_{i}\in \left\{0,1\right\}$ are referred as the *encryption keys*, and for any operator *U* we define $U^{0}=I$ and $U^{1}=U$. The subscript *i* in $X_{i},Z_{i}$, and $P_{i}$ is used to denote that the corresponding gate is applied on the *i*th qubit (hereinafter referred to as qubit *i*). Similarly, the subscript *i* in $a_{i},b_{i},c_{i}$ is used to denote that the corresponding encryption keys are related to qubit *i*. We can check that this encryption scheme is a quantum one-time pad (see Equation (Equation 11)), thus it provides an information-theoretical security for any qubit ρ.
(11)$$\frac{1}{4}\sum_{a,b,c\in \{0,1\}}X^{a}Z^{b}P^{c}\rho P^{3c}Z^{b}X^{a}=\frac{I}{2}.$$

In theory, to achieve this encryption, the client needs to perform random gates $P^{c}$, $Z^{b}$, and $X^{a}$ on the state ρ in sequence. However, for the qubit $|{+}_{x_{i}}\rangle$, it can be easily verified that
(12)$$X^{a_{i}}Z^{b_{i}}P^{c_{i}}|{+}_{x_{i}}\rangle \equiv |{+}_{\varphi_{i}}\rangle ,$$
where $\varphi_{i}={\left(-1\right)}^{a_{i}}\left(x_{i}+b_{i}+\frac{c_{i}}{2}\right)\pi \;\mathrm{mod}\;2\pi \in \left\{0,\frac{\pi }{2},\pi ,\frac{3\pi }{2}\right\}$. Thus, instead of preparing $|{+}_{x_{i}}\rangle$ then encrypting it by $X_{i}^{a_{i}}Z_{i}^{b_{i}}P_{i}^{c_{i}}$, the client can directly generate the encrypted qubit. Specifically, given the *i*th input bit $x_{i}\in \left\{0,1\right\}$, the client randomly chooses the corresponding encryption keys $a_{i},b_{i},c_{i}\in \left\{0,1\right\}$, then computes the value $\varphi_{i}={\left(-1\right)}^{a_{i}}\left(x_{i}+b_{i}+\frac{c_{i}}{2}\right)\pi \;\mathrm{mod}\;2\pi$. Finally, the client prepares the qubit $|{+}_{\varphi_{i}}\rangle$ as the encrypted qubit *i*.

After preparing all encrypted input qubits, the client sends them to the server. The server then performs the delegated quantum circuit *U* on the encrypted qubits. Here, the circuit *U* is known to both client and server (they can negotiate in advance via a classical channel). We assume that this circuit has been decomposed into a sequence of basic gates from the gate set U. That is, $U=U_{m}U_{m-1}\cdots U_{2}U_{1}$, where each $U_{i}\in U$ and the positive integer number *m* is the total number of gates. The following identities, which all hold up to an irrelevant global phase, can be easily verified.
(13)$$\begin{array}{c} X_{i}\left(X_{i}^{a_{i}}Z_{i}^{b_{i}}P_{i}^{c_{i}}\right)\equiv \left(X_{i}^{a_{i}}Z_{i}^{b_{i}⊕c_{i}}P_{i}^{c_{i}}\right)X_{i}, \end{array}$$
(14)$$\begin{array}{c} Z_{i}\left(X_{i}^{a_{i}}Z_{i}^{b_{i}}P_{i}^{c_{i}}\right)\equiv \left(X_{i}^{a_{i}}Z_{i}^{b_{i}}P_{i}^{c_{i}}\right)Z_{i}, \end{array}$$
(15)$$\begin{array}{c} P_{i}\left(X_{i}^{a_{i}}Z_{i}^{b_{i}}P_{i}^{c_{i}}\right)\equiv \left(X_{i}^{a_{i}}Z_{i}^{a_{i}⊕b_{i}}P_{i}^{c_{i}}\right)P_{i}, \end{array}$$
(16)$$\begin{array}{c} T_{i}\left(X_{i}^{a_{i}}Z_{i}^{b_{i}}P_{i}^{c_{i}}\right)\equiv \left(X_{i}^{a_{i}}Z_{i}^{a_{i}⊕b_{i}⊕\left(a_{i}c_{i}\right)}P_{i}^{a_{i}⊕c_{i}}\right)T_{i}, \end{array}$$
(17)$$\begin{array}{c} CZ_{i,j}\left(X_{i}^{a_{i}}Z_{i}^{b_{i}}P_{i}^{c_{i}}X_{j}^{a_{j}}Z_{j}^{b_{j}}P_{j}^{c_{j}}\right)\equiv \left(X_{i}^{a_{i}}Z_{i}^{a_{j}⊕b_{i}}P_{i}^{c_{i}}X_{j}^{a_{j}}Z_{j}^{a_{i}⊕b_{j}}P_{j}^{c_{j}}\right)CZ_{i,j}, \end{array}$$

It follows from Equations (Equation 13)–(Equation 17) that the basic gates X,Z,P,T,CZ are *commutable* with the encryption operator $X^{a}Z^{b}P^{c}$, although the encryption keys may need to be updated. For example, Equation (Equation 13) indicates that performing an $X_{i}^{a_{i}}Z_{i}^{b_{i}}P_{i}^{c_{i}}$ followed by an $X_{i}$ is equivalent to performing an $X_{i}$ followed by an $X_{i}^{a_{i}}Z_{i}^{b_{i}⊕c_{i}}P_{i}^{c_{i}}$. Thus, the client only needs to update the value of $b_{i}$ such that $b_{i}:=b_{i}⊕c_{i}$. The cases for $Z_{i},P_{i},T_{i}$, and $CZ_{i,j}$ follow the same reason. The related updating rules of encryption keys are shown in Equations (Equation 14)–(Equation 17). Note, however, that the commutativity noted above is not suited for the Hadamard gate *H*, as there is no $HP^{c}\equiv P^{c^{\prime }}H$ for any $c,c^{\prime }\in \left\{0,1\right\}$. In [38], the authors proposed a quantum teleportation scheme that they called the *H*-gadget (see Figure 1) so as to implement the *H* gate in a similar manner. Specifically, the client needs to prepare two ancillary qubits $|{+}_{\alpha_{i}}\rangle ,|{+}_{\beta_{i}}\rangle$ and a measurement angle $\theta_{i}$, where $\alpha_{i}$ and $\beta_{i}$ are chosen randomly, whereas $\theta_{i}$ can be determined by the following way.

Note that for any $\alpha_{i},\beta_{i}\in \left\{0,\frac{\pi }{2},\pi ,\frac{3\pi }{2}\right\}$, we can express them uniquely as follows: (18)$$\alpha_{i}=\left(d_{i}+\frac{e_{i}}{2}\right)\pi ,\beta_{i}=\left(f_{i}+\frac{g_{i}}{2}\right)\pi ,$$
where $d_{i},e_{i},f_{i},g_{i}\in \left\{0,1\right\}$. Thus, the client can first generate random bits $d_{i},e_{i},f_{i},g_{i}$ then compute the values of $\alpha_{i}$ and $\beta_{i}$. To determine $\theta_{i}$, the client generates a random bit, denoted by $h_{i}\in \{0,1\}$, then computes $\theta_{i}$ such that
(19)$$\theta_{i}=\left[h_{i}⊕b_{i}⊕d_{i}⊕\left(a_{i}c_{i}\right)⊕\left(s_{i}c_{i}\right)⊕\left(c_{i}e_{i}\right)\right]\pi +\frac{c_{i}⊕e_{i}}{2}\pi .$$

Note also that $\theta_{i}$ is relevant to the measurement outcome $s_{i}$, which means it can be determined until the client obtains the first measurement outcome $s_{i}$ from the server. Nevertheless, in theory, all qubits including ancillary qubits can be sent to the server before the computation begins. Thus, the complete procedure is classically interactive. Finally, the updating rule for *H* is shown as follows: (20)$$a_{i}^{\prime }=s_{i}^{\prime }⊕h_{i},b_{i}^{\prime }=a_{i}⊕s_{i}⊕f_{i}⊕\left[g_{i}\left(s_{i}^{\prime }⊕h_{i}\right)\right],c_{i}^{\prime }=g_{i},$$
where $a_{i}^{\prime },b_{i}^{\prime },c_{i}^{\prime }$ denote the updated encryption keys related to qubit *i*. The correctness of the *H*-gadget is given in the Appendix A. The detailed security proof of the protocol can be found in [38].

## 4. Distributed Architecture for Secure Delegated Quantum Computations

In this section, we give a simple scheme to implement the non-local CZ gate between two quantum servers. Our method uses the entangled state |H〉 (see Equation (Equation 10) for its definition) as ancillary qubits. The similar schemes have been studied intensively, for example, in [39,40]. The basic circuit is shown in Figure 2a. In the following content, we first verify the circuit identity shown in Figure 2, then, based on this circuit identity, we construct a distributed architecture for secure delegated quantum computations.

We start with a circuit named *X-teleportation* [40] (see Figure 3a), which is easy to verify.

First, we substitute a CZ and two *H* gates for the CX gate, obtaining the equivalent circuit, as shown in Figure 3b. We then convert the measurement basis from *Z* to *X* by the following identity (see Figure 4), which is also easy to verify. Finally, we obtain a variant of the *X*-teleportation that consists of H,CZ, and *X*-basis measurement, as shown in Figure 5.

We now turn back to Figure 2a. Note first that the CZ gate commutes with itself, thus the circuit can be reorganized, as in Figure 6a. Obviously, the partial circuits in the red-dotted line and blue-dotted line boxes are exactly the same circuit as the one in Figure 5, where X=M(0). Therefore, we can see that, after measuring qubits i,j, the rest qubits and the rest CZ gate comprise the circuit as, in Figure 6b. Finally, we use the following identity to exchange the positions of *X* and CZ, which can be easily verified: (21)$$CZ\cdot \left(X^{s}⊗I\right)=\left(X^{s}⊗Z^{s}\right)\cdot CZ,$$
where s∈{0,1}. Substituting the above identity in Figure 6b and considering the symmetry of CZ gate, we immediately obtain the desired circuit, as shown in Figure 2b.

Considering the encryption operators $X_{i}^{a_{i}}Z_{i}^{b_{i}}P_{i}^{c_{i}}$ and $X_{j}^{a_{j}}Z_{j}^{b_{j}}P_{j}^{c_{j}}$ on qubits *i* and *j*, we can see from Figure 6b that the non-local CZ can be thought to be performed on qubits i,j, which are encrypted by $X_{i}^{a_{i}⊕s_{i}}Z_{i}^{b_{i}}P_{i}^{c_{i}}$ and $X_{j}^{a_{j}⊕s_{j}}Z_{j}^{b_{j}}P_{j}^{c_{j}}$, thus according to the updating rule shown in Equation (Equation 17), we immediately obtain the updating rule of the non-local CZ gate as follows: (22)$$\left\{\begin{array}{c} a_{i}^{\prime }=a_{i}⊕s_{i},\\ b_{i}^{\prime }=a_{j}⊕s_{j}⊕b_{i},\\ c_{i}^{\prime }=c_{i}, \end{array}\right.\left\{\begin{array}{c} a_{j}^{\prime }=a_{j}⊕s_{j},\\ b_{j}^{\prime }=a_{i}⊕s_{i}⊕b_{j},\\ c_{j}^{\prime }=c_{j}. \end{array}\right.$$

Based on the above analysis, we construct a distributed architecture for secure delegated quantum computation, where a classical client equipped with some qubit generator can delegate an *n*-qubit circuit to *d* small-scale quantum servers. Without loss of generality, we assume that n=dk. In this configuration, each server typically needs a 2k-qubit register to process *k* input qubits of the *n*-qubit circuit. That is, for each qubit in the *n*-qubit circuit, the server needs a 2-qubit register to simulate it. To make sure 2k<n, it requires that d>2. We show this distributed architecture in Figure 7. Note that there is a special third party in this distributed architecture, which is used to generate and distribute entangled states |H〉 between all quantum servers. Thus, all servers do not need to be interconnected directly by a quantum (even classical) channel, as there is no information exchange between servers during the computation.

We give the complete procedure of the protocol in terms of pseudo-code (see Algorithms  1–3). For simplicity, we use C and ${\left\{S_{q}\right\}}_{q=1}^{d}$ to denote the client and *d* servers, respectively. That is, the *q*th quantum server is referred to as $S_{q}$. As noted, each server only processes *k* input qubits of the *n*-qubit delegated circuit. More specifically, for $S_{q}$, it only processes the qubits indexed by (q−1)k+1,(q−1)k+2,⋯,qk. Thus, in the case of no confusion, we also use $S_{q}=\left\{\left(q-1\right)k+1,\left(q-1\right)k+2,\cdots ,qk\right\}$ to denote the corresponding qubits. In addition, the delegated circuit *U* is formally expressed as $U=U_{m}^{p_{m}}U_{m-1}^{p_{m-1}}\cdots U_{1}^{p_{1}}$, where $p_{i}\subset \left\{1,2,\cdots ,n\right\}$ denotes the qubits on which the basic gate $U_{i}$ is exerted. For example, if $U_{i}^{p_{i}}$ is a CZ gate on qubits *k* and *l*, then $p_{i}=\left\{k,l\right\}$. By this definition, we can see that there must be $p_{i}\subset S_{q}$ if $U_{i}^{p_{i}}$ is a local gate in $S_{q}$, otherwise it only can be $p_{i}\subset S_{q}\cup S_{q^{\prime }}$ for some $S_{q}$ and $S_{q^{\prime }}$.
**Algorithm 1** Distributed Secure Delegated Quantum Computations**Input:** $x=x_{1}x_{2}\cdots x_{n}$                  // private against all $S_{q}$$U=U_{m}^{p_{m}}U_{m-1}^{p_{m-1}}\cdots U_{1}^{p_{1}}$         // public for C and all $S_{q}$**Output:** $y=y_{1}y_{2}\cdots y_{n}$                // private against all $S_{q}$1:C generates $a,b,c\leftarrow_{R}{\left\{0,1\right\}}^{n}$ and computes rotation angles $\left(\varphi_{1},\cdots ,\varphi_{n}\right)$ according to Equation (Equation 12), then prepares $|{+}_{\varphi_{1}}\rangle \cdots |{+}_{\varphi_{n}}\rangle$ as the encrypted input state, finally sends the qubits (q−1)k+1,q(k−1)+2,⋯,qk to $S_{q}$ where q=1,2,⋯,d. Specifically, C sends the qubits 1,2,⋯,k to $S_{1}$ then sends the qubits k+1,k+2,⋯,2k to $S_{2}$, and so on2:**for**i←1,m**do**3: **if** $U_{i}^{p_{i}}\in \left\{X,Z,P,T,H\right\}$ and $p_{i}\subset S_{q}$ for some q∈{1,2,⋯,d} **then**4:  **if** $U_{i}^{p_{i}}$ is not *H* **then**5:   $S_{q}$ performs $U_{i}^{p_{i}}$ on qubit $p_{i}$ while C updates the encryption keys of this qubit according to the updating rules shown in Equations (Equation 13)–(Equation 16)6:  **else**7:   C calls the **procedure**
Hadamard($p_{i},q$) (See Algorithm 2)8:  **end if**9: **else**           //
$U_{i}^{p_{i}}$ is a CZ gate on qubits $p_{i}$10:  **if** $p_{i}\subset S_{q}$ for some q∈{1,2,⋯,d} **then**11:   $S_{q}$ performs $U_{i}^{p_{i}}$ on qubits $p_{i}$ while C updates the encryption keys of those qubits according to the updating rule shown in Equation (Equation 17)12:  **else**        //
$p_{i}\subset S_{q}\cup S_{q^{\prime }}$ for some $q,q^{\prime }\in \left\{1,2,\cdots ,d\right\}$13:   C calls the **procedure**
Nonlocal-CZ($p_{i},q,q^{\prime }$) (See Algorithm 3)14:  **end if**15: **end if**16:**end for**17:Each server measures the final *k* qubits in *Z* basis, then sends the measurement outcomes to C         // let $\tilde{y}\in {\left\{0,1\right\}}^{n}$ be the result collected from all servers18:C computes the output $y=\tilde{y}⊕a$.         //
*a* is the *X* encryption keys of the final state

**Algorithm 2** Implement an *H* gate on qubit *i* where *i* is in $S_{q}$
1:**procedure**Hadamard(i,q)    // qubit *i* is encrypted by $X^{a_{i}}Z^{b_{i}}P^{c_{i}}$2: C generates $d_{i},e_{i}\leftarrow_{R}\left\{0,1\right\}$ and computes the angle $\alpha_{i}$ according to Equation (Equation 18), then prepares and sends the ancillary qubit $|{+}_{\alpha_{i}}\rangle$ to $S_{q}$3: $S_{q}$ performs $H_{i}$ and CZ gates on qubit *i* and $|{+}_{\alpha_{i}}\rangle$, then measures qubit *i* and sends the measurement outcome $s_{i}$ to C, finally labels the ancillary qubit as *i*4: C generates $f_{i},g_{i},h_{i}\leftarrow_{R}\left\{0,1\right\}$ and computes the angles $\beta_{i}$ and $\theta_{i}$ according to Equations (Equation 18) and (Equation 19), respectively, then prepares the ancillary qubit $|{+}_{\beta_{i}}\rangle$ and sends it with $\theta_{i}$ to $S_{q}$5: $S_{q}$ performs a CZ gate on qubit *i* and $|{+}_{\beta_{i}}\rangle$, then measures qubit *i* with $M(\theta_{i})$ and sends the measurement outcome $s_{i}^{\prime }$ to C, finally labels the ancillary qubit as *i*6: C updates the encryption keys of qubit *i* according to Equation (Equation 20)7:
**end procedure**



**Algorithm 3** Implement a nonlocal CZ gate on qubits *i* and *j* where *i* is in $S_{q}$ while *j* is in $S_{q^{\prime }}$, that is, $\left\{i,j\right\}\subset S_{q}\cup S_{q^{\prime }}$
1:**procedure**Nonlocal-CZ($\left\{i,j\right\},q,q^{\prime }$)   // qubits *i* and *j* are encrypted by $X^{a_{i}}Z^{b_{i}}P^{c_{i}}$ and $X^{a_{j}}Z^{b_{j}}P^{c_{j}}$, respectively2: C delegates the third party to prepare an entangled state |H〉 and distribute it to $S_{q}$ and $S_{q^{\prime }}$, that is, each server holds one qubit of |H〉 as the ancillary qubit3: $S_{q}$ ($S_{q^{\prime }}$) performs $H_{i}$ ($H_{j}$) and CZ gates on qubit *i* (*j*) and its ancillary qubit, then measures qubit *i* (*j*) and sends the measurement outcome $s_{i}$ ($s_{j}$) to C, finally labels its ancillary qubit as *i* (*j*)4: C updates the encryption keys of qubits *i* and *j* according to Equation (Equation 22)5:
**end procedure**



## 5. The Security of the Distributed Delegated Quantum Computation

We show that our protocol can guarantee the unconditional privacy of the input and output of the computation. We only consider that all servers and the third party who serves as an entanglement resource are *honest-but-curious*, which means they follow the algorithm honestly but try to obtain the information about the input and output. For example, they may record all classical information generated during the computation and cooperate with each other, even with the third party.

For the input, the conclusion is obvious as the client encrypts each input qubit by a quantum one-time pad. Therefore, to complete the proof, we only need to prove that the output state of the computation is also encrypted by a *unbiased* quantum one-time pad. In other words, there is no information leakage about the encryption keys during the computation. From the procedures of Algorithm 1, we can see that only when the client calls the **procedures** Hadamard and Nonlocal-CZ will there be an interaction between client and servers. In the other cases, the algorithm is non-interactive, which means there is no information leakage about the encryption keys from client to server as they do not exchange any information. Based on this observation, we infer that to prove the privacy we only need to analyze the procedures that implement the *H* and the nonlocal CZ gates.

We first consider the **procedure** Hadamard(i,q). In the following content, we use S to denote all servers including the *untrusted* third party. According to Algorithm 2, we can see that given the qubit *i* encrypted by $X^{a_{i}}Z^{b_{i}}P^{c_{i}}$ where $i\subset S_{q}$, S controls two ancillary qubits $Z^{d_{i}}P^{e_{i}}|+\rangle$ and $Z^{f_{i}}P^{g_{i}}|+\rangle$, and receives a measurement angle $\theta_{i}$ from C, it also generates two measurement outcomes $s_{i},s_{i}^{\prime }\in \left\{0,1\right\}$ from two independent measurements. We can infer from the below state evolution that the measurement outcomes $s_{i},s_{i}^{\prime }$ are uniformly random, thus S can obtain no information gain about any encryption keys according to $s_{i}$ and $s_{i}^{\prime }$.
(23)$$|\phi \rangle |+\rangle \overset{H⊗I}{\to }\left(H|\phi \rangle \right)|+\rangle \overset{CZ}{\to }\frac{|+\rangle }{\sqrt{2}}|\phi \rangle +\frac{|-\rangle }{\sqrt{2}}X|\phi \rangle .$$

The only available information to S now is the measurement angle $\theta_{i}$. Let $\theta_{i}$ be $u_{i}\pi +\frac{v_{i}\pi }{2}$, where $u_{i},v_{i}\in \left\{0,1\right\}$, then according to Equation (Equation 19), we know that $u_{i}$ and $v_{i}$ can be expressed as follows:
(24a)$$\begin{array}{cc} \; & u_{i}=h_{i}⊕b_{i}⊕d_{i}⊕\left(a_{i}c_{i}\right)⊕\left(s_{i}c_{i}\right)⊕\left(c_{i}e_{i}\right), \end{array}$$
(24b)$$\begin{array}{cc} \; & \;\;\;v_{i}=c_{i}⊕e_{i}, \end{array}$$
where $u_{i},v_{i}$, and $s_{i}$ are known to S. Intuitively, given $u_{i},v_{i}$, and $s_{i}$, no server can determine the correct values of $a_{i},b_{i},c_{i},d_{i},e_{i},h_{i}$, as there are six variables in two equations. Nevertheless, S may gain some information utilizing $u_{i}$ and $v_{i}$. For example, if $v_{i}=1$, then S can infer that $c_{i}e_{i}=0$. Substituting this into Equation ([Disp-formula FD24a-entropy-24-00794]), S can obtain a simplified equality $u_{i}=h_{i}⊕b_{i}⊕d_{i}⊕\left(a_{i}⊕s_{i}\right)c_{i}$. Despite this fact, we can show that there is no information leakage about all variables from $a_{i}$ to $h_{i}$. That is, we prove that in the view of S, the following equality holds true: (25)$$\mathrm{Pr}\left[r_{i}|u_{i},v_{i}\right]=\mathrm{Pr}\left[r_{i}\right]=\frac{1}{2},$$
where the random variable $r_{i}$ represents the possible parameters $\left\{a_{i},b_{i},c_{i},d_{i},e_{i},f_{i},g_{i},h_{i}\right\}$. To see that, we need to know the following simple facts.

First, if x,y∈{0,1} and *x* is uniform, i.e., $x\in_{R}\left\{0,1\right\}$, then x⊕y is also uniform. Second, if x,y∈{0,1} are uniform and let z=x⊕y, then Pr[x|z]=Pr[x]=1/2. Finally, if $x,y_{1},y_{2}\in \left\{0,1\right\}$ and *x* is uniform, let $z=x⊕(y_{1}y_{2})$, then $\mathrm{Pr}\left[y_{1}|z\right]=\mathrm{Pr}\left[y_{1}\right]$. These three basic facts can be easily verified. With these facts, we can complete our proof. Define $\xi_{i}=b_{i}⊕d_{i}⊕\left(a_{i}c_{i}\right)⊕\left(s_{i}c_{i}\right)⊕\left(c_{i}e_{i}\right)$ so that $u_{i}=h_{i}⊕\xi_{i}$. As $b_{i},d_{i}\in_{R}\left\{0,1\right\}$, we first know that $\xi_{i}\in_{R}\left\{0,1\right\}$. Furthermore, as $h_{i},\xi_{i}\in_{R}\left\{0,1\right\}$, we can get that $\mathrm{Pr}\left[h_{i}|u_{i}\right]=\mathrm{Pr}\left[h_{i}\right]=1/2$. Likewise, we can also get $\mathrm{Pr}\left[b_{i}|u_{i}\right]=\mathrm{Pr}\left[b_{i}\right]=1/2$ and $\mathrm{Pr}\left[d_{i}|u_{i}\right]=\mathrm{Pr}\left[d_{i}\right]=1/2$. For $a_{i}\in_{R}\left\{0,1\right\}$, define $\xi_{i}=h_{i}⊕b_{i}⊕d_{i}⊕\left(s_{i}c_{i}\right)⊕\left(c_{i}e_{i}\right)$ so that $u_{i}=\xi_{i}⊕\left(a_{i}c_{i}\right)$, from which we can infer that $\mathrm{Pr}\left[a_{i}|u_{i}\right]=\mathrm{Pr}\left[a_{i}\right]=1/2$. Note that $h_{i},b_{i},d_{i}$, and $a_{i}$ are irrelevant to $v_{i}$, which means $\mathrm{Pr}\left[r_{i}|u_{i},v_{i}\right]=\mathrm{Pr}\left[r_{i}|u_{i}\right]$ for any $r_{i}\in \left\{h_{i},b_{i},d_{i},a_{i}\right\}$. As for $c_{i},e_{i}\in_{R}\left\{0,1\right\}$, as they are related to both $u_{i}$ and $v_{i}$, in order to simplify our analysis, we define $h_{i}^{\prime }=h_{i}⊕\left(a_{i}c_{i}\right)$, $b_{i}^{\prime }=b_{i}⊕\left(s_{i}c_{i}\right)$, and $d_{i}^{\prime }=d_{i}⊕\left(c_{i}e_{i}\right)$, then obtain that $u_{i}=h_{i}^{\prime }⊕b_{i}^{\prime }⊕d_{i}^{\prime }$. Clearly, $h_{i}^{\prime },b_{i}^{\prime },d_{i}^{\prime }\in_{R}\left\{0,1\right\}$, so $c_{i}$ and $e_{i}$ are only related to $v_{i}$. By this, we can easily get that $\mathrm{Pr}\left[c_{i}|u_{i},v_{i}\right]=\mathrm{Pr}\left[c_{i}|v_{i}\right]=\mathrm{Pr}\left[c_{i}\right]=1/2$ and $\mathrm{Pr}\left[e_{i}|u_{i},v_{i}\right]=\mathrm{Pr}\left[e_{i}|v_{i}\right]=\mathrm{Pr}\left[e_{i}\right]=1/2$. Finally, $f_{i}$ and $g_{i}\in_{R}\left\{0,1\right\}$ are obviously irrelevant to $u_{i}$ and $v_{i}$ (see Equations ([Disp-formula FD24a-entropy-24-00794]) and ([Disp-formula FD24b-entropy-24-00794])), which means $\mathrm{Pr}\left[f_{i}|u_{i},v_{i}\right]=\mathrm{Pr}\left[f_{i}\right]=1/2$ and $\mathrm{Pr}\left[g_{i}|u_{i},v_{i}\right]=\mathrm{Pr}\left[g_{i}\right]=1/2$. So far, we have proved the statement in Equation (Equation 25), from which we know that the servers can obtain no information gain about $a_{i},b_{i},c_{i},d_{i},e_{i},f_{i},g_{i},h_{i}$ from the $\theta_{i}$. Thus, after the **procedure**
Hadamard(i,q), the updated keys $a_{i}^{\prime },b_{i}^{\prime },c_{i}^{\prime }$ are also secure.

Finally, we consider the **procedure** Nonlocal-CZ($\left\{i,j\right\},q,q^{\prime }$), where $\left\{i,j\right\}\in S_{q}\cup S_{q^{\prime }}$. Note that in this procedure, S can only obtain two independent and uniform measurement outcomes $s_{i},s_{j}$. According to the updating rules shown in Equation (Equation 22), we can see that as long as the encryption keys $\left\{a_{i},b_{i},c_{i}\right\}$ and $\left\{a_{j},b_{j},c_{j}\right\}$ are secure then the updated keys will also be secure against the servers. As a result, we conclude that, from the perspective of all servers, the output state of the computation is still encrypted by a sound quantum one-time pad.

## 6. Discussion

In this work, we proposed a secure distributed delegated quantum computation protocol, which allows clients to delegate their private computation to several quantum servers. We have shown that unconditional security of the input and output of the computation can be guaranteed as long as all servers follow the protocol honestly. Nevertheless, there are some notable problems in our work when we consider it in practice. In the end of this paper, we discuss those practical problems.

First, note that our protocol can only work well in a noise-free environment. To make our protocol fault-tolerant, we assume that each quantum server must be capable of performing *fault-tolerant quantum computation* [41]. However, this would inevitably increase the overhead of ancillary qubits. In addition, we need to consider two channel noises: one is between the client and each server, the other is between the third party and each server. The former will introduce errors in the input state, whereas the latter will introduce errors in the entangled state. There are some methods to remedy this problem. For the input state, the client can utilize some *quantum error-correct code* [42] to protect each qubit. However, it requires that the client can perform additional quantum operations. As for the entangled state, each pair of servers can use some *quantum entanglement distill* [43] protocol to obtain the entangled states with high fidelity. Similarly, it requires additional local operations and classical communications between the servers.

Second, note that our protocol can only protect the security of the input and output of the computation. This is because the model of the delegated quantum computation we used in our work is SDQC protocol instead of UBQC protocol. Nevertheless, we can convert, in principle, a SDQC protocol into a UBQC protocol. To do that, we first encode the delegated circuit *U* as a binary string denoted by C(U). Next, according to the quantum computation theory [44], there exists a universal quantum circuit U such that
(26)$$U|{+}_{C(U)}\rangle |{+}_{x}\rangle =|{+}_{C(U)}\rangle U|{+}_{x}\rangle ,$$
where the input of the universal circuit U consists of two parts: $|{+}_{x}\rangle$ is the input state of *U* and $|{+}_{C(U)}\rangle$ is the canonical and quantum description of the circuit *U*. Performing this universal circuit U in our protocol, we can apparently achieve a blind distributed delegated quantum computation.

Last, we should note that in this work we only consider the honest servers and the third party who perform the protocol as the client desires. However, a real server may not follow the protocol honestly, and an untrusted third party may prepare some other entangled states for the servers. To detect such a malicious server including the untrusted third party, we should introduce a verification mechanics in our protocol. Indeed, verification is an important topic in the quantum computation theory (see [45,46]). There is an easy way to achieve the verification in our protocol. Specifically, given the delegated circuit *U*, the client can introduce another small quantum circuit *V*, for example, a permutation circuit [47], which is easy to simulate on a classical computer. The client then randomly inserts the qubits of *V* into the circuit *U* and runs this hybrid circuit on the universal quantum circuit U. After the computation, the client check the result of *V*; if the result does not match the desired, then the client rejects the output.

## Figures and Tables

**Figure 1 entropy-24-00794-f001:**
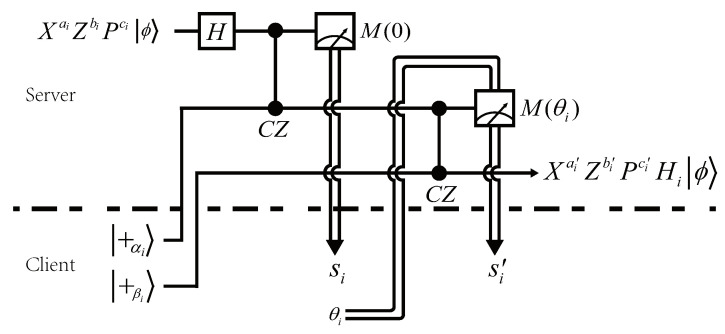
The *H*-gadget in Ref. [38], which is designed for implementing an *H* gate on an encrypted qubit *i*, where $s_{i},s_{i}^{\prime }\in \left\{0,1\right\}$ are the measurement outcomes and $\alpha_{i},\beta_{i}\in \left\{0,\frac{\pi }{2},\pi ,\frac{3\pi }{2}\right\}$ are the rotation angles of two ancillary qubits, and $\theta_{i}\in \left\{0,\frac{\pi }{2},\pi ,\frac{3\pi }{2}\right\}$ is the measurement angle of the second measurement.

**Figure 2 entropy-24-00794-f002:**
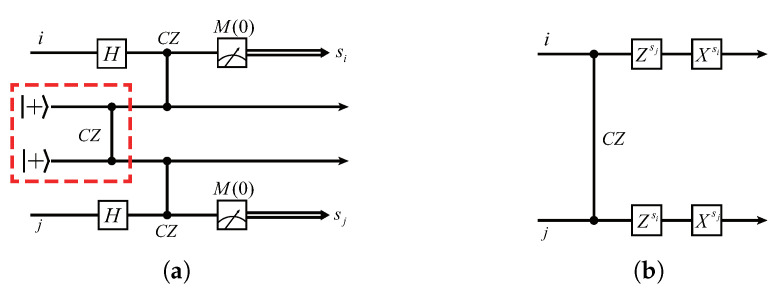
(**a**) The basic circuit used to implement a non-local CZ gate on two distant qubits *i* and *j*, where the partial circuit in the red dotted box is used to generate the entangled state |H〉. (**b**) The equivalent quantum circuit for (**a**).

**Figure 3 entropy-24-00794-f003:**
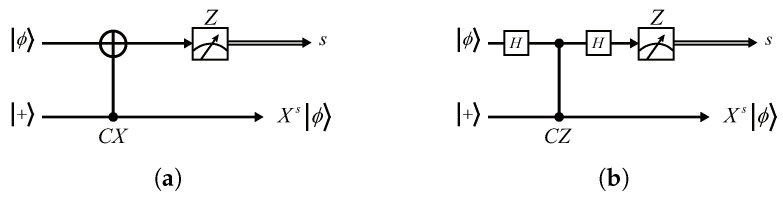
(**a**) The original *X*-teleportation in [40]; (**b**) the *X*-teleportation that replaces the CX with a CZ and two *H* gates. In both circuits, the measurement is performed under *Z* basis.

**Figure 4 entropy-24-00794-f004:**
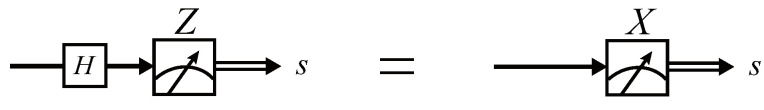
Measurement identity that converts *Z*-basis to *X*-basis.

**Figure 5 entropy-24-00794-f005:**
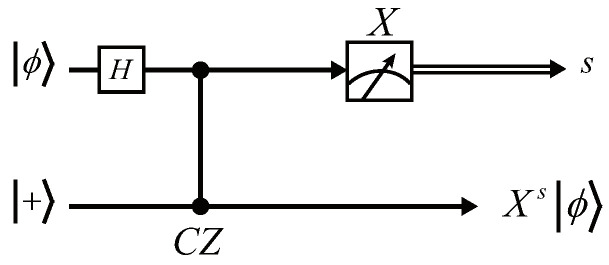
The variant *X*-teleportation consisting of CZ and *H* gates, where the measurement basis is *X*.

**Figure 6 entropy-24-00794-f006:**
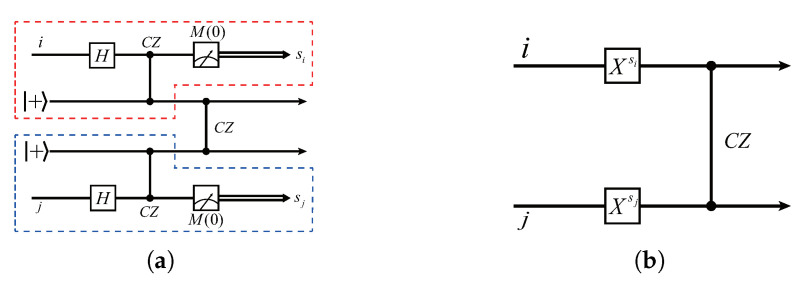
(**a**) The equivalent form of the circuit shown in Figure 2a. (**b**) The resulting circuit after measuring qubits i,j.

**Figure 7 entropy-24-00794-f007:**
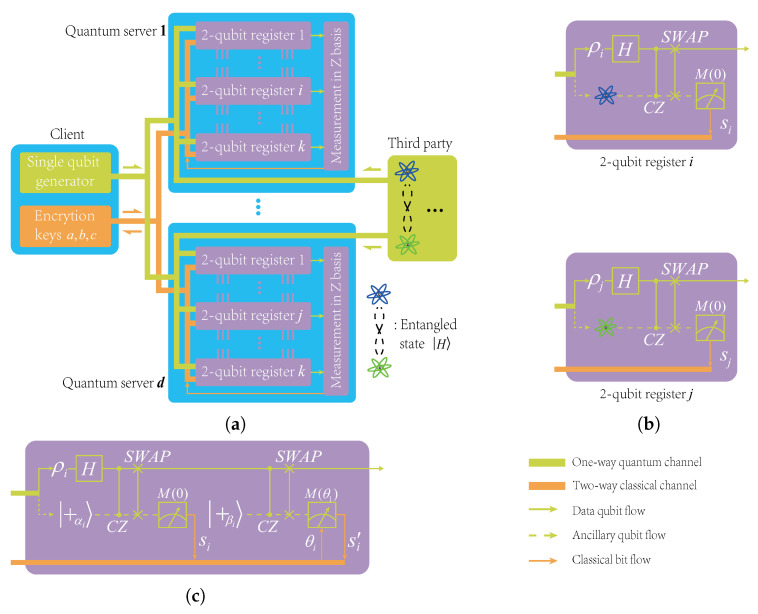
(**a**) The distributed architecture for secure delegated quantum computations; (**b**) the circuits for a CZ gate between two nonlocal registers *i* and *j*; (**c**) the circuit for an *H* gate in any register *i*.

## Data Availability

Not applicable.

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
