# Peer review of "A Distributed Architecture for Secure Delegated Quantum Computation"

_entropy, 2022, doi:10.3390/e24060794_

Round 1
Reviewer 1 Report
In this work, the authors define a distributed secure delegated quantum computation protocol.
The manuscript is well written and technically sound. After some improvements I can recommend it for publication.
- Some variables are not defined carefully in the current version, please check all definitions.
- I suggest to add a brief section on the application of the results via a near-term gate-model quantum computer setting, see also the suggested references
- I suggest to update the list of references with the following items:
[1] Quantum supremacy using a programmable superconducting processor, Nature,
Vol 574, DOI:10.1038/s41586-019-1666-5 (2019).
[2] Quantum Computing in the NISQ era and beyond, Quantum 2, 79 (2018).
[3] Quantum Computational Supremacy, Nature, vol 549, pages 203-209 (2017).
Reviewer 2 Report
Please see the attachment.

Reviewer 3 Report
The paper introduces an exciting protocol where entangled servers perform delegated quantum computation. The security analysis considers the honest-but-curious case, resulting in a simple analysis. Still, it would be much more interesting to continue with dishonest servers or a minority of dishonest servers. Nevertheless, I do think that the weaker security scenario analysis should award a publication anyway,
The issues that I would like to have clarified are the following:
Is there any advantage in having entangled qubits instead of flying photons? It is known that a quantum channel can be emulated using quantum teleportation by using EPR pairs together with a classical channel.
What is the difference between your protocol and one where we adopt a delegated Quantum Computation using quantum teleportation instead of flying qubits?
Moreover, is there any advantage in using them instead of a quantum channel? One needs the quantum channel to send the EPR. Moreover, storing them without noise might be a problem. Can a clear advantage be more specific?
So do you assume that the EPR generator can generate other states, or are they honest but curious as well? Because in the abstract of the manuscript this is not clear, it seems that the EPR generator is untrusted, but in the proof, he can only generate EPR pairs. I do not see how can the EPR generator be used to be an honest-but-curious attacker; he can only generate EPR pairs and nothing else; there is no attack scenario here (can you confirm this?). It would be much more interesting if a malicious server could use the state generated from the server to gain some information.
The English is quite good
